# NeIn: Telling What You Don't Want

Nhat-Tan Bui[1], Dinh-Hieu Hoang[2], Quoc-Huy Trinh[3,4], Minh-Triet Tran[2], Truong Nguyen[5], Susan Gauch[1]

[1]University of Arkansas, USA   [2]University of Science, VNU-HCM, Vietnam

[3]Aalto University, Finland   [4]SpexAI GmbH, Germany   [5]University of California, San Diego, USA

https://tanbuinhat.github.io/NeIn/

## Abstract

*Negation is a fundamental linguistic concept used by humans to convey information that they do not desire. Despite this, minimal research has focused on negation within text-guided image editing. This lack of research means that vision-language models (VLMs) for image editing may struggle to understand negation, implying that they struggle to provide accurate results. One barrier to achieving human-level intelligence is the lack of a standard collection by which research into negation can be evaluated. This paper presents the first large-scale dataset, Negative Instruction (NeIn), for studying negation within instruction-based image editing. Our dataset comprises 366,957 quintuplets, i.e., source image, original caption, selected object, negative sentence, and target image in total, including 342,775 queries for training and 24,182 queries for benchmarking image editing methods. Specifically, we automatically generate NeIn based on a large, existing vision-language dataset, MS-COCO, via two steps: generation and filtering. During the generation phase, we leverage two VLMs, BLIP and InstructPix2Pix (fine-tuned on MagicBrush dataset), to generate NeIn's samples and the negative clauses that expresses the content of the source image. In the subsequent filtering phase, we apply BLIP and LLaVA-NeXT to remove erroneous samples. Additionally, we introduce an evaluation protocol to assess the negation understanding for image editing models. Extensive experiments using our dataset across multiple VLMs for text-guided image editing demonstrate that even recent state-of-the-art VLMs struggle to understand negative queries.*

## 1. Introduction

When it comes to training vision-language models (VLMs), we have to consider a wide range of human information needs, requiring systems to handle a wide range of user-generated queries. They range from simple and straightforward ones like "describe the image" to complex prompts involving rich contextual detail and creative reasoning.

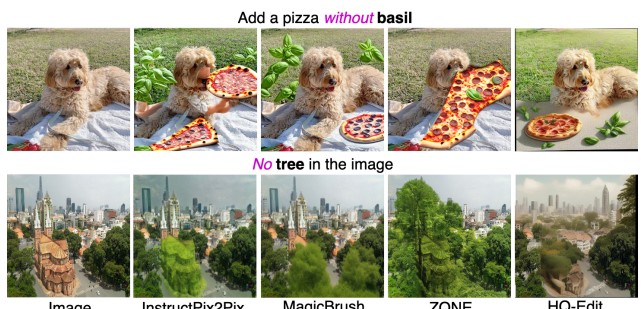

Figure 1. The failures of recent text-guided image editing methods in understanding the negative queries.

Inspired by Laurence R. Horn [7], *"negation is a sine qua non of every human language but is absent from otherwise complex systems of animal communication."* In this work, we address the problem of negative queries that specify information that should be excluded, a ubiquitous feature in human language. Examples of negative queries in image editing tasks include "The bathroom area *without* a curtain" or "The street with a person, but *not* with a car."

Research that explicitly tackles the negation problem for neural networks has mainly focused on natural language understanding [14, 20, 25, 27, 30], and a few vision-language tasks including video retrieval [26]. However, in image editing, many recent SOTAs such as InstructPix2Pix [1], MagicBrush [28], ZONE [10], and HQ-Edit [8] fail to understanding negative queries, as Figure 1 illustrates. In contrast, the models seem to focus on adding objects that need to be excluded from the input images (for the query *no tree in the image*, more trees are added, occluding the cathedral).

Although image editing VLMs can be prompted with instructions such as "remove," understanding negation is crucial for these models to achieve human-level intelligence. This is because humans frequently use negation in various ways, and not every negative cue can simply be replaced by using "remove."

One possible reason why VLMs fail to understand negation is the lack of negative descriptions in current image-caption pair datasets, e.g., MS-COCO [11], SBU Captions [17], CC12M [3], LAION-400M [22], etc. Since the na-

| Datasets | Tasks | Train | | Validation | | Total | |
|---|---|---|---|---|---|---|---|
| | | #Negative | #All | #Negative | #All | #Negative | #All |
| CC12M [3] | Pre-training | – | – | – | – | 314,181 (2.53%) | 12,423,374 |
| LAION-400M [22] | Pre-training | – | – | – | – | 2,404,784 (0.58%) | 413,862,224 |
| MS-COCO'14 [11] | Image Captioning | 1,761 (0.43%) | 414,113 | 886 (0.44%) | 202,654 | – | 616,767 |
| SBU Captions [17] | Image Captioning | – | – | – | – | 26,222 (2.62%) | 1,000,000 |
| CC3M [23] | Image Captioning | 54,219 (1.63%) | 3,318,333 | – | – | – | 3,369,218 |
| CIRR [13] | Composed Image Retrieval | 868 (3.08%) | 28,225 | 130 (3.11%) | 4,181 | – | 36,554 |
| InstructPix2Pix[1] | Image Editing | 77 (0.02%) | 313,010 | – | – | – | 313,010 |
| MagicBrush [28] | Image Editing | 54 (0.61%) | 8,807 | 6 (1.17%) | 528 | – | 10,388 |

Table 1. Statistic of captions in current image-caption pair datasets. We identify negative captions by 32 negative words: "no", "not", "without", "don't", "doesn't", "never", "none", "neither", "nothing", "can't", "isn't", "aren't", "didn't", "did not", "isn't", "is not", "aren't", "are not", "wasn't", "was not", "weren't", "were not", "won't", "will not", "hasn't", "has not", "haven't", "have not", "can't", "can not", "couldn't", and "could not".

ture of captions in these datasets is to describe objects and visual concepts in the image as well as the stories related to them, they lack negative clauses. As illustrated in Table 1, the number of negative sentences in image-caption pair datasets is very small. Furthermore, some captions contain negative words but they actually do not describe what is not present in the image, e.g., "do not spend money in stores with this sign," "things you cannot miss." This leads to two consequences: (1) VLMs do not understand the meaning of negative words because of the heavily-biased dataset during training, and (2) there is no evaluation data to assess the capability of VLMs in understanding negation.

To tackle this issue, we present a pipeline for constructing a new dataset, **Ne**gative **In**struction (NeIn[1]), designed for training and evaluating VLMs on negation understanding, specifically within the context of image editing. Particularly, we use BLIP [9] to identify objects that are not present in the source image. We expand the captions that represent the content of source image by incorporating negative words for objects that are not present. One example of a negative sentence would be "The image *doesn't* have an *apple*". We then generate counter-example target images that contain those absent objects using InstructPix2Pix [1] fine-tuned on MagicBrush [28]. We retain only acceptable-quality images by filtering target samples using BLIP and LLaVA-NeXT [12] to remove images that have been excessively altered from the original content of the source image or that make the excluded object unrecognizable. Thus, by combining the target NeIn sample with the corresponding negative query, we can obtain the source image.

In order to evaluate the performance of VLMs on our dataset, we propose an evaluation method for image editing models to assess both the ability to remove objects from images in response to negative clauses and the ability to retain the original objects in the image after modification by the queries. To summarize, our main contributions are:
- We investigate the ability of VLMs to interpret negation cues in text-guided image editing, leading to the creation

of the first large-scale vision-language negation dataset for this task, termed NeIn.
- We introduce a pipeline to generate NeIn, an extensive dataset comprising 366,957 quintuplets. This dataset focuses on the understanding of negation, a fundamental linguistic concept, for image editing VLMs.
- We propose an evaluation method for negation understanding that can be used by future researchers. Using our evaluation method, we observe that VLMs in image editing task have difficulty comprehending negative instructions. This insight opens a new research direction for improving negation understanding for VLMs.

## 2. Related Work

**Negation Understanding**, a major linguistic topic, is becoming prominent in research. In natural language understanding, Ravichander et al. proposed CondaQA [20] which is a question answering dataset specifically designed for negation understanding. Experiments conducted on CondaQA revealed that deep learning models may have a simple trick that they reverse the rank list when they see the negation cues, leading to acceptable results when encountering fully negated queries. Nevertheless, their performance still suffer a severe degradation on composed queries. Truong et al. [25] investigated the LLMs' ability to understand negation under various settings. NevIR [27] benchmarked models against a simple yet brilliant task, ranking two paragraphs regarding a question that is relevant to only one of them. Measured in pair-wise accuracy, volunteers easily achieved a perfect score of 100%, far superior to most of the models whose performances are below 25%. The best ones, using cross encoders, scored below 50%. ExcluIR [30] included an evaluation benchmark comprising 3,452 manually curated queries, along with a training set of 70,293 queries with a positive document and a negative document. The authors have concluded that even after fine-tuned on negated dataset, all models still lag behind human performance a great deal. SetBERT [14] recently proposed fine-tuning on a synthesized dataset with a focus on pre-

---
[1]Nein means "no" in German.

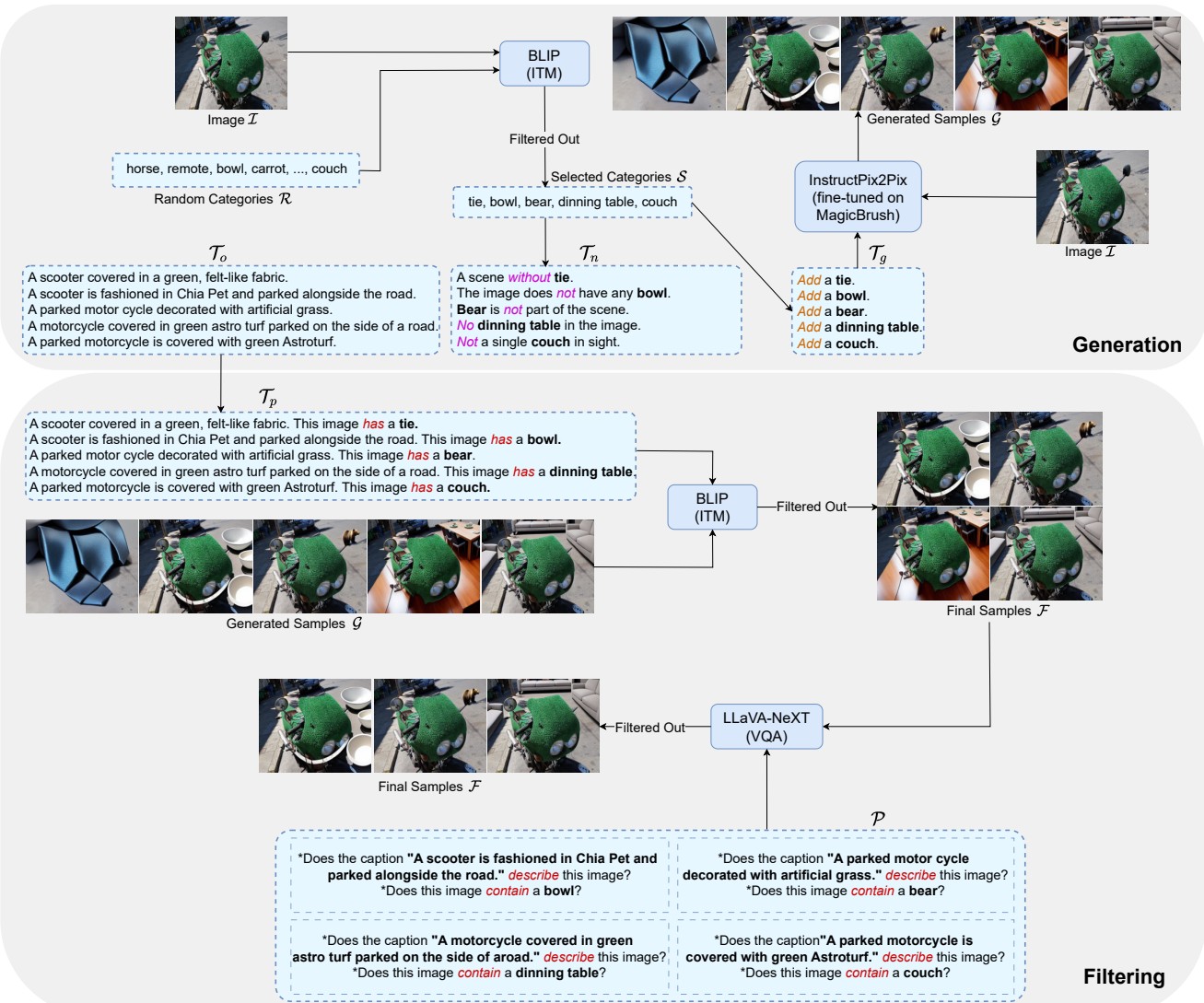

Figure 2. The process to create our dataset. It consists of two main steps: generation and filtering. ITM and VQA are image-text matching and visual question answering, respectively.

dicting negative samples using inverse-contrastive learning, while BoolAttn [15] took a different approach by directly adjusting attention scores to downweight tokens affected by negation. In conclusion, all the results unanimously show that even SOTAs struggle to comprehend negation cues.

For visual data retrieval, Wang et al. [26] measured models on original dataset MSR-VTT3k, its negated version, and composed version. However, models can cheat on the negated version by inverting the result on the original dataset ($\Delta$Recall and $\Delta$MIR are nearly zero) but on composed queries, all the models perform very poorly (Recalls at rank N=1, 10 are respectively less than 12% and 46%, and Mean Inverted Rank is less than 0.23), lagging behind their own performance on the original dataset by a large margin.

Similar to our work, Singh et al. [24] recently intro-

duced the CC-Neg dataset tailored for negation problem, contains 228,246 images, true captions and negated captions. However, the negated captions of CC-Neg falsely describe their corresponding images. More precisely, the authors generated the negated captions so that they and their images can be used as negative pairs in the contrastive loss to finetune the vision-language pre-trained models, whereas in our dataset NeIn, the negated captions still describe exactly their original images. Other differences will be clarified in the Sections 3 and 4.

## 3. NeIn Dataset

NeIn is designed specifically to address the challenge of negation understanding in VLMs for image editing. Each

| 1) The image *doesn't* have any $\{\mathcal{S}\}$. | 2) $\{\mathcal{S}\}$ is *not* part of the scene. | 3) *No* $\{\mathcal{S}\}$ present in the image. | 4) The image is *without* $\{\mathcal{S}\}$. |
|---|---|---|---|
| 5) The image does *not* have any $\{\mathcal{S}\}$. | 6) The image *lacks* $\{\mathcal{S}\}$. | 7) *No* $\{\mathcal{S}\}$ in the image. | 8) A scene *without* $\{\mathcal{S}\}$. |
| 9) The image *cannot* have any $\{\mathcal{S}\}$. | 10) *Not* a single $\{\mathcal{S}\}$ in sight. | 11) $\{\mathcal{S}\}$ is *missing* from the image. | 12) The image *lacks* the presence of $\{\mathcal{S}\}$. |
| 13) $\{\mathcal{S}\}$ is *nowhere* to be seen in the image. | | | |

Table 2. The pre-defined formats for $\mathcal{T}_n$. These formats contain the negative words: "no", "not", "without", "doesn't", "does not", "cannot", "lacks", "missing", and "nowhere".

tuple in NeIn consists of a natural image $\mathcal{I}$ and its original caption $\mathcal{T}_o$, selected object $\mathcal{S}$, sentence containing negative clause $\mathcal{T}_n$ and a synthetic image $\mathcal{G}$ that satisfies the original caption but not its corresponding negative sentence.

The creation of NeIn involves two primary stages: the first stage is *generation*, which employs BLIP [9] and InstructPix2Pix [1], fine-tuned on MagicBrush[28], to generate target samples; the second stage is *filtering*, where BLIP and LLaVA-NeXT [12] are utilized to remove erroneous samples.

### 3.1. Generation Pipeline

The data generation process of NeIn is shown in upper part of Figure 2. The main idea is that given image $\mathcal{I}$ and a corresponding caption $\mathcal{T}_o$ describing what objects are present in $\mathcal{I}$, we will find a negative clause, termed $\mathcal{T}_n$, such that it satisfies the content of source image $\mathcal{I}$. In our case, $\mathcal{I}$ and $\mathcal{T}_o$ are the image and its corresponding captions from MS-COCO [11]. More precisely, for each tuple, we arbitrarily select a collection of 15 object categories $\mathcal{R}$ from MS-COCO. This means the categories vary for each tuple. Then, we find out which objects in $\mathcal{R}$ do not appear in $I$ using the image-text matching (ITM) from BLIP [9]. Note that the original caption $\mathcal{T}_o$ only describes the main objects and their relationships within the source image $\mathcal{I}$; it does not account for all the categories present in $\mathcal{I}$. For instance, $\mathcal{T}_o$ is "bedroom scene with a bookcase, blue comforter, and window," but $\mathcal{I}$ also includes a mirror, trees, a lamp, and a cabinet. Therefore, using BLIP instead of identifying objects in $\mathcal{T}_o$ to evaluate $\mathcal{R}$ will reduce errors for NeIn's samples. We choose BLIP over other VLMs because it is sufficiently fast to process large quantities of images. We accept false positive results (when an object is present in the image but BLIP fails to detect it) during the generation step. If an object is too small or distorted to the point where BLIP cannot recognize it, we consider that object is not present in the image.

We select 5 objects with lowest scores and generate their corresponding negative sentences based on some predefined formats, illustrated in Table 2. The negative sentences include negative words such as "doesn't," "not," "without" and $\mathcal{S}$ represents the selected objects, indicating that these objects are not present in the image $\mathcal{I}$. Concatenating $\mathcal{T}_n$ with the original caption $\mathcal{T}_o$, we attain a new caption that still matches the original image $\mathcal{I}$.

**Algorithm 1** Generation

**Input:**
$\mathcal{I}$: source image
$\mathcal{R}$: list of 15 random object categories
$\mathcal{P}$: list of pre-defined formats for negative clauses
**Output:**
$\mathcal{S}, \mathcal{T}_n, \mathcal{T}_g, \mathcal{G}$: lists of selected object categories, negative sentences, prompts, and their corresponding synthetic images, respectively

1: $\mathcal{S} := [], \mathcal{T}_n := [], \mathcal{T}_g := [], \mathcal{G} := []$
2: **for** each object $\mathcal{R}^{(i)}$ in $\mathcal{R}$ **do**          ▷ Categories not in $I$
3:     $c^{(i)} \leftarrow \text{ITM}(\mathcal{I}, \mathcal{R}^{(i)})$          ▷ Cosine similarity
       # threshold $\alpha = 0.4$
4:     **if** $c^{(i)} < \alpha$ **then**          ▷ Add categories
5:         append $(\mathcal{R}^{(i)}, c^{(i)})$ to $\mathcal{S}$
6:     **end if**
7: **end for**
8: $\mathcal{S} \leftarrow \text{top}(\mathcal{S}, 5)$     ▷ Select 5 lowest-cosine categories
9: **for** each object $\mathcal{S}^{(i)}$ in $\mathcal{S}$ **do**          ▷ Form sentences
10:     $p \leftarrow \text{random}(\mathcal{P})$          ▷ Choose a random format
11:     $\mathcal{T}_n^{(i)} \leftarrow \text{gen}(p, \mathcal{S}^{(i)})$          ▷ Negative sentence
12:     append $\mathcal{T}_n^{(i)}$ to $\mathcal{T}_n$
13:     $\mathcal{T}_g^{(i)} \leftarrow$ "Add a/an $\{\mathcal{S}^{(i)}\}$"     ▷ Instruction sentence
14:     append $\mathcal{T}_g^{(i)}$ to $\mathcal{T}_g$
15: **end for**
16: **for** each prompt $\mathcal{T}_g^{(i)}$ in $\mathcal{T}_g$ **do**          ▷ NeIn samples gen
17:     $\mathcal{G}^{(i)} \leftarrow \text{generator}(\mathcal{I}, \mathcal{T}_g^{(i)})$          ▷ Generated sample
18:     append $\mathcal{G}^{(i)}$ to $\mathcal{G}$
19: **end for**
20: **return** $\mathcal{S}, \mathcal{T}_n, \mathcal{T}_g, \mathcal{G}$

Next, our goal is to create an image $\mathcal{G}$ that $\mathcal{T}_o$ matches it but not $\mathcal{T}_n$, which means the object specified in $\mathcal{T}_n$ is present in $\mathcal{G}$. Theoretically, we can create $\mathcal{G}$ by any image editing method adding those objects to the source image. In the case of NeIn, we choose a version of InstructPix2Pix [1], a diffusion-based deep neural network, fine-tuned on MagicBrush [28] due to the high quality of its outputs. Thus, in the context of image editing, given image $\mathcal{G}$, $\mathcal{T}_n$ will be a query for removing some object in $\mathcal{G}$, taking $\mathcal{I}$ as one of the best results. For instance, if $\mathcal{I}$ is a picture of a dog in a garden, $\mathcal{T}_n$ could be "the photo does not have any laptop," and $\mathcal{G}$ would be a picture of a dog and a laptop in the garden. To summarize, Algorithm 1 illustrates the steps to accomplish the generation phase. Note that each image in MS-COCO

has 5 captions, so we can generate 5 tuples for each image.

## 3.2. Filtering Pipeline

The data filtering process of NeIn is shown in lower part of Figure 2. The main purpose of this stage is to eliminate images that significantly alter the content of query image $\mathcal{I}$ or difficultly identify object categories $\mathcal{S}$. The specific examples are the $1^{st}$ and $4^{th}$ images in $\mathcal{G}$ (Figure 2), where the image is completely transformed and no longer retains the original objects like the motorcycle and the green fabric. If the original content is no longer preserved or the object $S$ is hard to distinguish, then, when combined with $\mathcal{T}_n$, the model can no longer output $\mathcal{I}$.

We employ a two-stage filtering strategy: the first stage leverages the image-text matching (ITM) function from BLIP [9], while the second utilizes visual question answering (VQA) from LLaVA-NeXT [12]. We select these models because we need to verify two aspects: original captions and selected object categories. Using ITM (represented by BLIP) and VQA (represented by LLaVA-NeXT) enable us to design prompts for checking these two aspects.

In order to perform the first filtering stage, we generate $\mathcal{T}_p$ by combining $\mathcal{T}_o$ with a prompt "This image *has* a/an $\{\mathcal{S}\}$", where $\mathcal{S}$ represents the selected object category. We utilize BLIP to calculate the matching score of $\mathcal{T}_p$ and the generated samples $\mathcal{G}$ to eliminate the erroneous samples from $\mathcal{F}$.

Results from the first stage are further filtered in the second stage to ensure the quality of NeIn's samples. For this second stage, we design two prompts $\mathcal{P}$: (1) Does the caption $\mathcal{T}_o$ *describe* this image? and (2) Does this image *contain* a/an $\{\mathcal{S}\}$? The first prompt ensures that the content of the samples $\mathcal{F}$ does not deviate significantly from $\mathcal{I}$, while the second prompt checks whether the selected objects appear in $\mathcal{F}$. We require LLaVA-NeXT to output "Yes" or "No" for both prompts, and if either prompt returns "No," we remove that sample from $\mathcal{F}$. Algorithm 2 shows the pseudo code for our filtering phase. The examples of NeIn's sample after filtering step and the data statistics are shown in the supplementary material.

## 3.3. Discussion

First, in contrast to CC-Neg [24] whose tuples contain only a natural image, its corresponding caption, and a negated caption, we design NeIn so that for every caption, there is at least one target image associated with it. As a consequence, NeIn, with *a source image, corresponding captions, selected objects, negative sentences, and target images*, may be suitable for tasks that require target images (e.g., image editing, composed image retrieval, visual grounding), thus potentially having more applications than CC-Neg. Besides, CC-Neg only consists of basic negative words like "no", "not", "without"; while NeIn is more diverse.

---

**Algorithm 2** Filtering

**Input:**
$\mathcal{G}, \mathcal{T}_o$: generated images and its original caption
$\mathcal{S}$: list of selected object categories
**Output:**
$\mathcal{F}$: final samples of NeIn
1:  $\mathcal{F} \coloneqq []$
    # first filtering stage by ITM
2:  **for** each tuple $(\mathcal{S}^{(i)}, \mathcal{T}_o^{(i)}, \mathcal{G}^{(i)})$ in $(\mathcal{S}, \mathcal{T}_o, \mathcal{G})$ **do**
3:     $\mathcal{T}_p^{(i)} \leftarrow \mathcal{T}_o^{(i)}$ + "This image has a/an $\{\mathcal{S}^{(i)}\}$"
4:     $c^{(i)} \leftarrow \text{ITM}(\mathcal{G}^{(i)}, \mathcal{T}_p^{(i)})$     ▷ Cosine similarity
    # threshold $\alpha = 0.4$
5:     **if** $c^{(i)} > \alpha$ **then**     ▷ Add samples
6:         append $\mathcal{G}^{(i)}$ to $\mathcal{F}$
7:     **end if**
8:  **end for**
    # second filtering stage by VQA
9:  **for** each tuple $(\mathcal{S}^{(i)}, \mathcal{T}_o^{(i)}, \mathcal{F}^{(i)})$ in $(\mathcal{S}, \mathcal{T}_o, \mathcal{F})$ **do**
    # two pre-defined prompts
10:    $\mathcal{P}_1^{(i)} \leftarrow$ "Does the caption $\{\mathcal{T}_o^{(i)}\}$ describe this image?"
11:    $\mathcal{P}_2^{(i)} \leftarrow$ "Does this image contain a/an $\{\mathcal{S}^{(i)}\}$?"
12:    $b_1^{(i)} \leftarrow \text{VQA}(\mathcal{F}^{(i)}, \mathcal{P}_1^{(i)})$     ▷ Boolean result
13:    $b_2^{(i)} \leftarrow \text{VQA}(\mathcal{F}^{(i)}, \mathcal{P}_2^{(i)})$     ▷ Boolean result
14:    **if** $b_1^{(i)} = $ "No" **or** $b_2^{(i)} = $ "No" **then**   ▷ Filter out
15:       remove $\mathcal{F}^{(i)}$ from $\mathcal{F}$
16:    **end if**
17:  **end for**
18:  **return** $\mathcal{F}$

---

Second, this dataset is intended to support research on negation understanding, as a purely mathematical logic, rather than generating realistic images, which is related to naturalness. That means, "a carrot within a traffic scene" may seem absurd but humans can reliably and effortlessly determine whether there is a carrot in the photo regardless of other aspects of the photo. Therefore, the fact that NeIn's samples are synthetic does not impact the overall quality of the dataset. In other words, we want to assess the models' ability to answer logical questions, unaffected by irrelevant factors such as naturalness, context, artistic style.

Third, one concern is raised by how we generate images corresponding to $\mathcal{G}$. Generated by the image editing diffusion model InstructPix2Pix, fine-tuned using MagicBrush, the synthetic images are unrealistic and they may be corrupted. However, to the best of our knowledge, there is no traditional technique that would allow us to automatically add objects into an image without altering its main content since the added object may cover some important existing objects in the image. In practice, we observe that image-editing deep neural networks try to keep the main portion of the input image unharmed. Thus, we only need to ap-

ply some automatic filters to get rid of poor quality images rather than employing a tedious and unscalable manual process. From an input image and a caption, it only takes an average of 9 seconds to generate the corresponding image with clauses and 3 second to assess it by two-stage filtering strategy on an A100 GPU. In total, we spend approximately 86 days to create NeIn with a single A100 GPU.

Fourth, our negative clauses $\mathcal{T}_n$ are descriptive prompts, raising another concerns about whether evaluating image editing models with descriptive prompts is fair. However, existing evidence suggests that image editing methods can effectively handle descriptive prompts. For instance, "in a race car video game" and "it is now midnight" in from InstructPix2Pix paper; "the dog is looking forward" and "she is now cutting up carrots" from MagicBrush dataset; and "the tarantula is given a glowing outline and the background is changed to a dramatic sunset with vibrant reds and purples" from HQ-Edit dataset. This demonstrates that image editing models must understand descriptive prompts, therefore, they must also understand negative cues, e.g. "the tree is *without* a candle."

## 4. Experiments on Image Editing Task

### 4.1. Experiment Setup

We benchmark our evaluation set in five SOTA image editing methods, including InstructPix2Pix [1], MagicBrush [28], ZONE [10], HQ-Edit [8], and InstructDiffusion [4]. We fine-tune InstructPix2Pix and MagicBrush in our training set with 8 epochs. All baseline details, including the versions of BLIP and LLaVA-NeXT, are provided in the supplementary material.

For each tuple $i^{th}$, these models take $\mathcal{F}^{(i)}$ as an input image and $\mathcal{T}_n^{(i)}$ as an instruction, and return the target image $\mathcal{T}^{(i)}$; where the original image $\mathcal{I}^{(i)}$ is the ground truth; this is illustrated in Figure 3. In fairness, we conduct experiments with the default settings for each model.

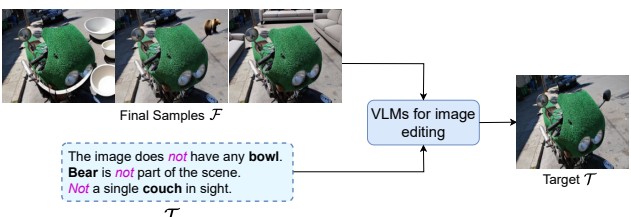

Figure 3. Illustration for fine-tuning and benchmarking process.

### 4.2. Evaluation Metrics

To assess the difference between the output of the image editing models $\mathcal{T}$ and ground truth $\mathcal{I}$, we consider two aspects: image quality and negative instruction satisfaction.

For image quality, we follow MagicBrush [28], employing four different metrics: L1, L2, CLIP-I, and DINO.

---

**Algorithm 3** Removal Evaluation by VQA

**Input:**
$\mathcal{T}$: considered model's outputs
$\mathcal{S}$: objects to be removed
**Output:**
$s$: removal score

1:   $s := 0$
2:   **for** each tuple $(\mathcal{T}^{(i)}, \mathcal{S}^{(i)})$ in $(\mathcal{T}, \mathcal{S})$ **do**
      # pre-defined prompt
3:      $p \leftarrow$ "Does this image contain a/an $\mathcal{S}^{(i)}$?"
4:      **if** VQA$(\mathcal{T}^{(i)}, p)$ = "No" **then**   ▷ Object is removed
5:         $s \leftarrow s + 1$
6:      **end if**
7:   **end for**
8:   $s \leftarrow s/|\mathcal{T}|$
9:   **return** $s$

---

The L1 and L2 metrics measure the pixel-level distance between $\mathcal{T}$ and $\mathcal{I}$. CLIP-I metric leverages CLIP [19] model, while DINO metric employs DINO [2] model to compute the alignment between target and query images by measuring the cosine similarity of their embeddings. We consider the image discrepancy in terms of realism using Clean-FID [18] metric, an improved version of the Frechet Inception Distance (FID) metric [5] that focuses on image resizing and quantization. In addition, human perceptual judgment is measured by LPIPS [29] score to more comprehensively evaluate $\mathcal{T}$ and $\mathcal{I}$ by considering human visual perception.

Then, to measure how much the output semantically satisfies the instruction, we consider whether image editing methods successfully eliminate the object categories specified in the negative sentence, and determine if these methods can preserve the objects not mentioned in the negative sentence. The first is determined by the *Removal score*, while the second is assessed using the *Retention score*.

Since the purpose of both metrics is to identify objects, we consider the visual question answering (VQA), represented by LLaVA-NeXT and open-vocabulary object detection (OVD), represented by OWLv2 [16]. Note that, different from the generation step, we do not accept false positive results for the evaluation metrics. Therefore, both LLaVA-NeXT and OWLv2 are suitable choices.

**Removal Evaluation.** The removal evaluation by VQA is illustrated by Algorithm 3, while the evaluation using OVD is in the supplementary. For VQA (LLaVA-NeXT), we prompt "Does this image *contain* a/an $\{\mathcal{S}^{(i)}\}$?" If the result is "No," the object is considered successfully removed. For OVD (OWLv2), we give the model output $\mathcal{T}^{(i)}$ and object $\mathcal{S}^{(i)}$. The result is a list of bounding boxes with confidence scores. The object is considered removed if the length of this list is zero. Addressing the concern that the bounding box may be misclassified, we calculate the Area Under the Curve (AUC) regarding the highest confidence score of

| Methods | Image Quality | | | | | | Negation Understanding | | | | |
|---|---|---|---|---|---|---|---|---|---|---|---|
| | | | | | | | LLaVA-NeXT | | OWLv2 | | |
| | L1 ↓ | L2 ↓ | CLIP-I ↑ | DINO ↑ | FID ↓ | LPIPS ↓ | Removal ↑ | Retention ↑ | Removal ↑ | AUC-Removal ↑ | Retention ↑ |
| InstructPix2Pix [1] | 11.24 | 3.59 | 81.68 | 73.53 | 10.60 | 0.43 | 3.83 | 81.96 | 6.70 | 50.11 | 81.63 |
| **InstructPix2Pix** | **8.32** | **2.32** | **93.11** | **91.67** | **4.08** | **0.33** | **93.62** | **98.26** | **92.66** | **97.89** | **95.83** |
| MagicBrush [28] | 8.95 | 2.69 | 88.29 | 84.91 | 7.80 | 0.36 | 5.06 | 93.86 | 8.13 | 52.48 | 91.39 |
| **MagicBrush** | **8.38** | **2.35** | **93.04** | **91.53** | **4.15** | **0.33** | **92.18** | **98.21** | **91.24** | **97.34** | **98.07** |
| ZONE [10] | 11.95 | 3.67 | 74.12 | 63.18 | 14.95 | 0.46 | 2.93 | 72.38 | 6.47 | 46.04 | 69.07 |
| HQ-Edit [8] | 23.48 | 9.61 | 62.84 | 46.60 | 27.61 | 0.67 | 32.23 | 54.75 | 40.42 | 70.29 | 57.43 |
| InstructDiffusion [4] | 8.54 | 2.54 | 90.57 | 88.62 | 6.89 | 0.34 | 31.46 | 97.55 | 30.00 | 67.99 | 97.58 |

Table 3. Quantitative results of five image editing SOTA methods on the evaluation set of NeIn. All the metrics are in (%). The Instruct-Pix2Pix and MagicBrush finetuned on NeIn's training set are highlighted. The FID used here is Clean-FID [18].

---

**Algorithm 4** Retention Evaluation by VQA

**Input:**

$\mathcal{F}$: samples of NeIn

$\mathcal{T}_o$: original caption from MS-COCO

$\mathcal{T}$: considered model's outputs

**Output:**

$s$: retention score

1: $s := 0$
2: **for** each tuple $(\mathcal{F}^{(i)}, \mathcal{T}_o^{(i)}, \mathcal{T}^{(i)})$ in $(\mathcal{F}, \mathcal{T}_o, \mathcal{T})$ **do**
3:    $\text{list}^1 := [], \text{list}^2 := []$
4:    $\mathcal{O} \leftarrow \text{extractor}(\mathcal{T}_o^{(i)})$    ▷ Original objects in $\mathcal{I}$
     # check $\mathcal{O}$ in $\mathcal{F}$
5:    **for** each $object$ in $\mathcal{O}$ **do**
6:      $p \leftarrow$ "Does this image contain a/an $object$ ?"
7:      $b \leftarrow \text{VQA}(\mathcal{F}^{(i)}, p)$    ▷ Boolean result
8:      **if** $b =$ "Yes" **then**    ▷ Object is still in $\mathcal{F}^{(i)}$
9:        append $object$ to $\text{list}^1$
10:      **end if**
11:    **end for**
     # check $\mathcal{O}$ in both $\mathcal{F}$ and $\mathcal{T}$
12:    **for** each $object$ in $\text{list}^1$ **do**
13:      $p \leftarrow$ "Does this image contain a/an $object$ ?"
14:      $b \leftarrow \text{VQA}(\mathcal{T}^{(i)}, p))$    ▷ Boolean result
15:      **if** $b =$ "Yes" **then**    ▷ Object is in $\mathcal{F}^{(i)}$ & $\mathcal{T}^{(i)}$
16:        append $object$ to $\text{list}^2$
17:      **end if**
18:    **end for**
19:    $score \leftarrow$ length of $\text{list}^2$ / length of $\text{list}^1$
20:    $s \leftarrow s + score$
21: **end for**
22: $s \leftarrow s / |\mathcal{T}|$
23: **return** $s$

---

each sample.

**Retention Evaluation.** We observe that, in the case of not being able to understand which object needs to be removed, the model may still achieve a high removal score by removing as many objects as possible from the images. Hence, we measure the retention score that assess whether or not the model retains the *salient* objects in the original images. Algorithm 4 shows the retention evaluation by VQA, with the corresponding evaluation for OVD provided

in the supplementary. Let's denote the original objects in $\mathcal{T}_o$ (i.e. objects have in source image $\mathcal{I}$) are $\mathcal{O}$. To split $\mathcal{O}$ from $\mathcal{T}_o$, we use Spacy [6], a popular library for Natural Language Processing. Note that we only consider the *main* objects, which are essential to the content of $\mathcal{I}$, therefore we use $\mathcal{T}_o$ because it covers the important objects in $\mathcal{I}$.

We first check that $\mathcal{O}$ is still present in the samples $\mathcal{F}$ of NeIn. If $\mathcal{O}$ exists in $\mathcal{F}$, we then check whether or not $\mathcal{O}$ is present in generated image $\mathcal{T}$ of considered model. We divide the number of retained objects by the number of $\mathcal{O}$ to get the retention score for each sample. The final retention score is the average across all samples of the evaluation set.

### 4.3. Results

**Quantitative Evaluation.** Table 3 presents the quantitative results between InstructPix2Pix, fine-tuned Instruct-Pix2Pix, MagicBrush, fine-tuned MagicBrush, ZONE, HQ-Edit, and InstructDiffusion on the evaluation set of NeIn.

None of the five methods perform well on pixel-level metrics, such as L1 and L2, or on image quality metrics, such as CLIP-I, DINO, FID, and LPIPS, indicating that negative prompts are considerably challenging. This is particularly evident when considering the Removal and Retention scores. Image editing models generally struggle to understand the meaning of negation when they do not remove the mentioned objects, as demonstrated by the low Removal, calculated by LLaVA-NeXT and OWLv2, and AUC-Removal scores of OWLv2. InstructPix2Pix [1], ZONE [10], and HQ-Edit [8] also distort the content of the source image, as can be seen by the low Retention score. It is noteworthy that HQ-Edit achieves better Removal score and AUC-Removal than the others but worse in retaining the original content of the images. We hypothesize that the model dramatically alters the images that makes their content indecipherable, as indicated by the high L1 and L2 scores. InstructDiffusion [4] handles negation best among the five baselines, likely due to its training on diverse computer vision datasets that enhance generalization for negative queries.

In contrast, the fine-tuned versions of InstructPix2Pix and MagicBrush significantly enhance both image quality and instruction satisfaction with negative queries. Given

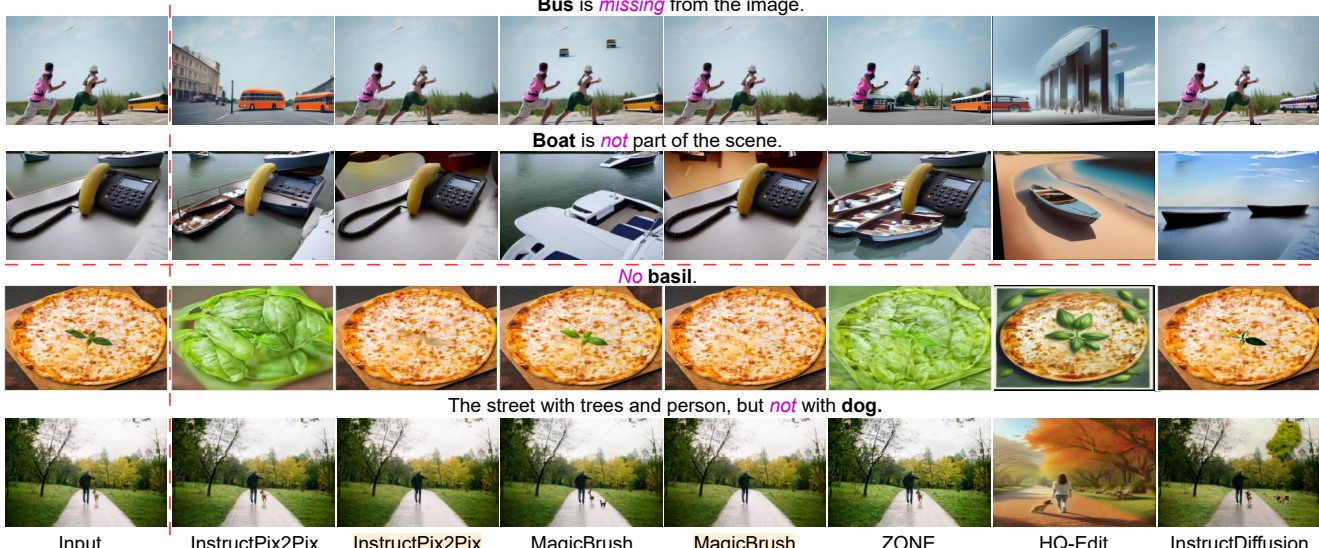

**Bus** is *missing* from the image.

**Boat** is *not* part of the scene.

*No* **basil**.

The street with trees and person, but *not* with **dog**.

| Input | InstructPix2Pix | InstructPix2Pix | MagicBrush | MagicBrush | ZONE | HQ-Edit | InstructDiffusion |

Figure 4. Qualitative results of five SOTA methods on NeIn's evaluation samples (first two samples) and random image-prompt pairs (last two samples). The fine-tuned InstructPix2Pix (3rd column) and MagicBrush (5th column) on NeIn's training set are highlighted.

that current image editing methods rely on Diffusion [21] framework. We assume by this evidence, other text-guided image editing methods are likely to achieve similar results.

**Qualitative Evaluation.** Some results for each image editing model are illustrated in Figure 4. We consider both image-prompt pairs from NeIn's samples and randomly pairs outside of NeIn's distribution to evaluate the generalization of fine-tuned versions.

Instead of removing the mentioned objects, original image editing models tend to have the following problems: (1) retaining the mentioned object in the edited image; (2) increasing the quantity of mentioned object in the generated image, and even bringing that object to the center of the images; and (3) completely replacing the content of the query image with that object. This observation demonstrates the failure of VLMs in image editing on negation understanding, potentially affecting other vision-language tasks.

On the contrary, the fine-tuned InstructPix2Pix and MagicBrush models clearly demonstrate the ability to remove objects specified in negative queries. Even when dealing with difficult prompt that include trees, person, and dog; models are still able to successfully understand negation. This suggests that, following fine-tuning with NeIn, VLMs may be capable of understanding negation.

## 5. Conclusion

We introduce NeIn, the first large-scale dataset for negation understanding within the context of text-guided image editing, comprising 366,957 quintuplets with 342,775 queries for training set and 24,182 queries for evaluation set. Negation understanding, an important linguistic concept yet to be fully explored in image editing, is a crucial task for aligning image editing VLMs with human information needs. We present a novel pipeline to automatically generate and filter samples for NeIn by leveraging VLMs for vision-language pre-training and image editing. Additionally, we present a comprehensive evaluation protocol, including removal and retention aspects, to assess the performance of current image editing models on negation understanding for NeIn's evaluation set. Our experiments reveal that existing image editing methods struggle to understand negative queries, highlighting a new challenge for the research community. By fine-tuning these models on NeIn's training set, we can improve their ability to identify negative terms in user queries, as demonstrated by both quantitative and qualitative results.

**Limitations.** Two current limitations of NeIn are that (1) we have only performed experiments using image editing models, and (2) the negative predefined prompts are relatively simple. **Future Directions.** Based on current limitations, future research plans to leverage and expand NeIn include: (1) fine-tuning and benchmarking NeIn for other tasks in vision-language domain such as composed image retrieval and image-text matching; (2) considering complex negative sentences involving words such as "except", "neither-nor", etc. We hope that with the release of NeIn, the research community will shift its focus toward negation understanding, which we believe is an important open problem for VLMs.

**Acknowledgement.** Nhat-Tan Bui is sponsored by the GPSC Tiffany Marcantonio Research Grant from the University of Arkansas. Minh-Triet Tran and Dinh-Hieu Hoang are funded by the University of Science, VNU-HCM, under grant number T2024-150.

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
