# NeIn: Telling What You Don't Want

## Supplementary Material

## A. Dataset Statistics

The MS-COCO [6] dataset includes a total of 123,287 images with 616,435 captions for training and evaluation. For each COCO caption, we generate a synthetic image for NeIn, resulting in 616,435 synthetic images in total. Following the first filtering stage using BLIP [4], NeIn contains a total of 530,694 samples, accounting for 86.1% of the dataset and excluding 85,741 (13.9%) erroneous entries. Subsequently, after filtering with LLaVA-NeXT [7], NeIn retains 366,957 samples (59.53% of the original set). In total, the two-stage filtering process eliminates 40.47% of erroneous samples. Figure 1(a) illustrates the number of samples after each filtering phase.

In the interest of efficiency, we randomly select 24,182 queries for benchmarking, while the remaining 342,775 queries are used for fine-tuning. This data split is suitable for image editing task, with approximately 6.6% for the evaluation set, since the evaluation set of MagicBrush accounts for 6%.

In order to exam the distribution within finalized dataset, i.e., after filtering process, we present the distribution of the pre-defined format for $\mathcal{T}_n$, as depicted in Figure 1(b).

We also analyze the number of instances per category for 80 object categories in Figure 1(c). Since NeIn is designed to address the problem of negation understanding, object imbalance does not affect the quality of our dataset. This imbalance phenomenon may be due to the difficulty of incorporating objects with low proportion in the distribution (e.g., stop sign, frisbee) into the context of COCO images using MagicBrush [9]. Consequently, generated images containing these objects are often filtered out by BLIP or LLaVA-NeXT.

## B. Dataset Format

Table 1 provides the format for one sample in the JSON file. Each sample in NeIn consists of five components: the source image, original caption, generated sentence, negative sentence, and the NeIn sample itself. These components are represented in the JSON file as "COCO" ($\mathcal{I}$), "T_original" ($\mathcal{T}_o$), "T_generated" ($\mathcal{T}_g$), "T_negative" ($\mathcal{T}_n$), and "NeIn" ($\mathcal{F}$), respectively.

For each caption in COCO, we can generate a corresponding sample for NeIn, the "NeIn_000000000074_5" indicates that this image is derived from the COCO image "COCO_000000000074," utilizing its fifth caption.

Based on the "NeIn" sample and the "T_negative" clause, the ground truth corresponds to the "COCO" image. The "T_original" from COCO is employed for the *retention*

evaluation, whereas the "T_generated" is utilized to extract the selected object category ($\mathcal{S}$) for the *removal* evaluation.

```
{
"COCO":  "COCO_000000000074",
"T_original":  "A puppy rests on
the street next to a bicycle.",
"T_generated":  "Add a couch.",
"T_negative":  "The image cannot
have any couch.",
"NeIn":  "NeIn_000000000074_5"
}
```

Table 1. The JSON format of NeIn.

## C. NeIn's Sample After Filtering Process

Figure 2 illustrates the examples of NeIn after the two-stage filtering process. In the first example, all samples are retained after filtering. This occurs when the final samples $\mathcal{F}$ pass both the BLIP and LLaVA-NeXT checks. In contrast, in the second example, the first, fourth, and fifth samples are rejected. The first and fourth samples are removed because the objects added by MagicBrush are not recognized as a hair dryer and bench by BLIP. In the case of the fifth sample, BLIP recognizes the presence of a boat, whereas LLaVA-NeXT does not assent its presence in the image. Using a two-stage filtering process with image-text matching (BLIP) and visual question answering (LLaVA-NeXT) ensures the high quality for NeIn's samples.

## D. Baseline Details

We leverage API from Transformers library[1] for both BLIP and LLaVA-NeXT. We use the large version for BLIP[2]. We select BLIP over more recent vision-language models for image-text matching in generation phase because it is trained on MS-COCO dataset, and the objects (e.g., "car," "apple") are relatively simple compared to complex texts that BLIP can perform. While for the filtering phase, BLIP is additionally supported by LLaVA-NeXT. Leveraging BLIP accelerates the process of generating datasets with a large number of samples. The threshold of 0.4 is selected based on experiments. For LLaVA-NeXT, we employ the Mistral-7B version[3] to strike a balance between accuracy and resource efficiency.

---

[1] https://huggingface.co/docs/transformers/index
[2] https://huggingface.co/Salesforce/blip-itm-large-coco
[3] https://huggingface.co/liuhaotian/llava-v1.6-mistral-7b

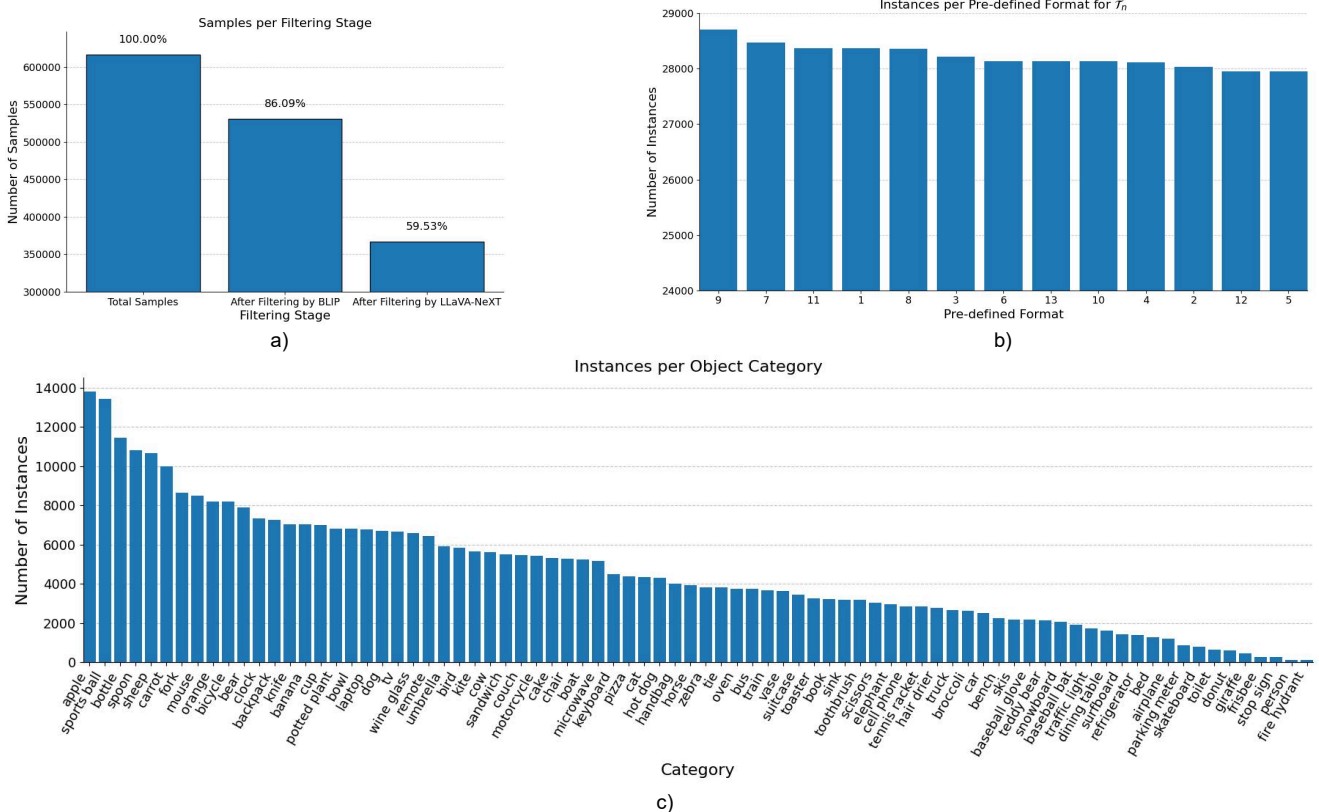

Figure 1. Statistical analysis for NeIn. a) Number of samples after each filtering phase; b) Number of instances per pre-defined format for $\mathcal{T}_n$, the x-axis is followed to the order presented Table 2; c) Number of instances per object category, these 80 objects follow the objects in MS-COCO dataset. Best view in zoom.

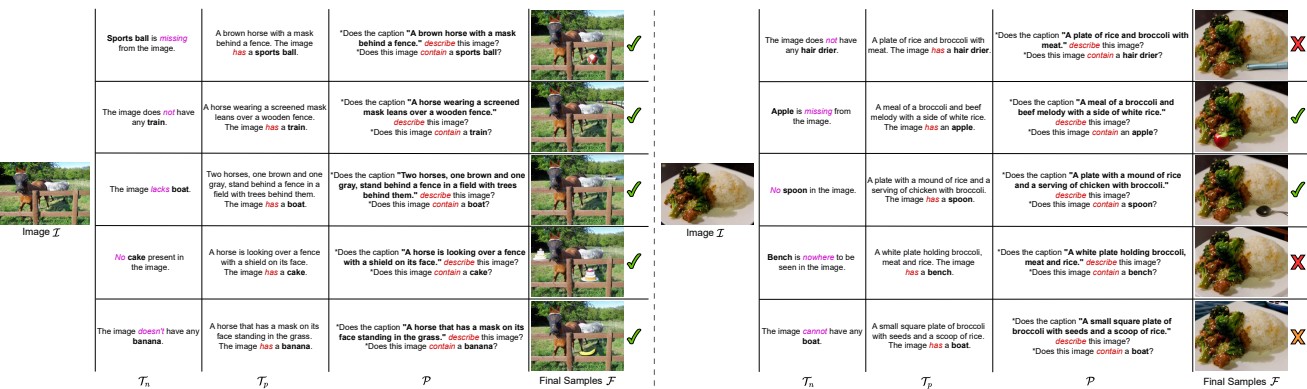

Figure 2. Two examples for our filtering process. The green ✓ indicates samples that are retained. The red ✗ signifies samples removed by BLIP, while the orange ✗ checkmark denotes samples removed by LLaVA-NeXT. Best view in zoom.

We evaluate NeIn on five SOTA methods published in 2023 and 2024: InstructPix2Pix [1][4], MagicBrush [9][5], ZONE [5][6], HQ-Edit [3][7], and InstructDiffusion [2][8]:

1) InstructPix2Pix proposes a multi-modal training dataset comprising input images, text-based editing instruc-

[4]https://github.com/timothybrooks/instruct-pix2pix
[5]https://github.com/OSU-NLP-Group/MagicBrush
[6]https://github.com/lsl001006/ZONE
[7]https://github.com/UCSC-VLAA/HQ-Edit
[8]https://github.com/cientgu/InstructDiffusion

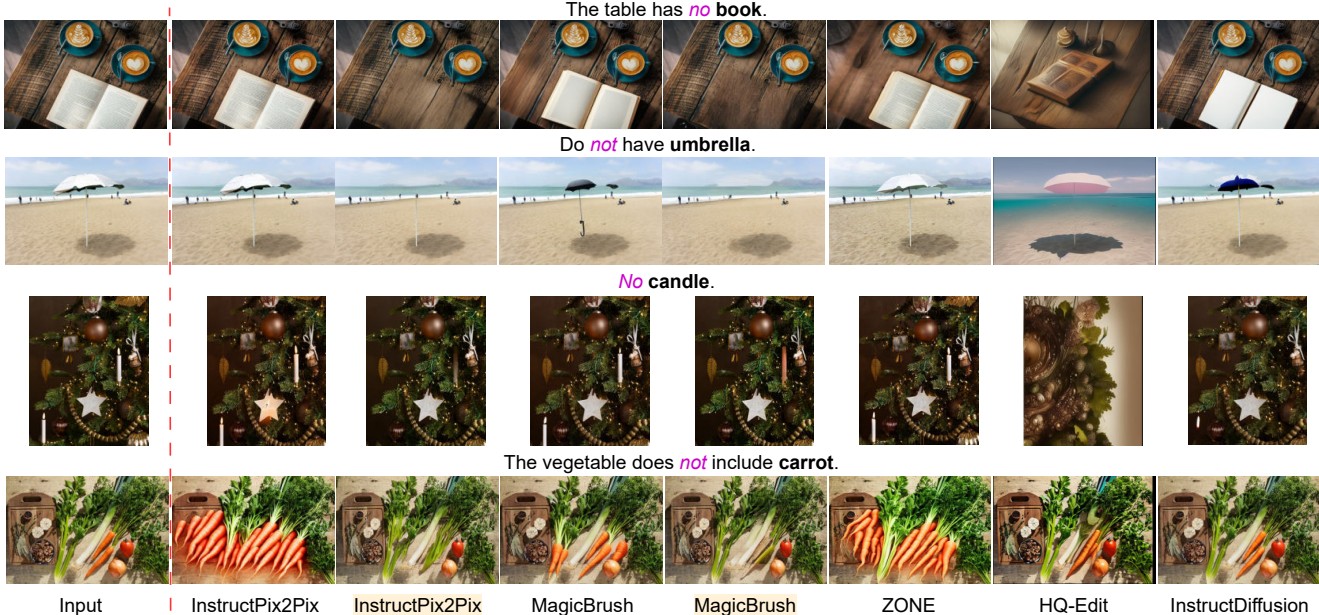

The table has *no* **book**.

Do *not* have **umbrella**.

*No* **candle**.

The vegetable does *not* include **carrot**.

| Input | InstructPix2Pix | InstructPix2Pix | MagicBrush | MagicBrush | ZONE | HQ-Edit | InstructDiffusion |

Figure 3. Additional qualitative results of five SOTA methods. The fine-tuned InstructPix2Pix (3rd column) and MagicBrush (5th column) on NeIn's training set are highlighted.

---

**Algorithm 1** Removal Evaluation by OVD

**Input:**
$\mathcal{T}$: considered model's outputs
$\mathcal{S}$: objects to be removed

**Output:**
$s$: removal score

1: $s := 0$
2: **for** each tuple $(\mathcal{T}^{(i)}, \mathcal{S}^{(i)})$ in $(\mathcal{T}, \mathcal{S})$ **do**
3:      $p \leftarrow \text{OVD}(\mathcal{T}^{(i)}, \mathcal{S}^{(i)})$        ▷ Prediction list
4:      **if** length of $p = 0$ **then**     ▷ Object is removed
5:          $s \leftarrow s + 1$
6:      **end if**
7: **end for**
8: $s \leftarrow s/|\mathcal{T}|$
9: **return** $s$

---

tions, and the corresponding edited images. It fine-tunes Stable Diffusion [8] by this dataset in a supervised manner to achieve zero-shot image editing.

2) MagicBrush is a manually annotated dataset with 10,388 samples covering both single- and multi-turn image editing scenarios. The fine-tuned version for Instruct-Pix2Pix demonstrates superior editing performance compared to other approaches.

3) ZONE initially utilizes InstructPix2Pix to identify the editing locations given the text instructions. It then refines those regions using the Region-IoU scheme combined with the Fourier-based edge smoother. ZONE produces high-quality results for intuitive instructions such as "add", "re-

move", and "change".

4) HQ-Edit introduces a new image editing dataset comprising approximately 200,000 edits, by leveraging the large language and text-to-image models. The fine-tuned version for InstructPix2Pix generates high-quality edited results, further validating its effectiveness for image editing.

5) InstructionDiffusion generalizes various computer vision tasks as instruction-based image editing. In order to achieve that goal, it combines multiple datasets for keypoint detection, semantic segmentation, referring segmentation, image enhancement (e.g., denoising, deblurring, and watermark removal), and image editing. The results indicate that the multi-task learning strategy is benefit for image editing.

## E. Removal and Retention Evaluation by Open-Vocabulary Detection (OVD)

Algorithm 1 and algorithm 2 illustrate the removal and retention evaluation by OVD (OWLv2). Instead of using predefined prompts as in VQA, we leverage the capabilities of an open-vocabulary object detection model, specifically OWLv2, to detect arbitrary objects in images.

## F. Additional Qualitative Results

We provide more qualitative results of five baselines and two fine-tuned versions for negative prompts in Figure 4. The original versions of the baseline models generally fail to recognize negative words. They tend to either retain the object in the image, replace the object with a similar one of a different appearance, add more objects to the image, or

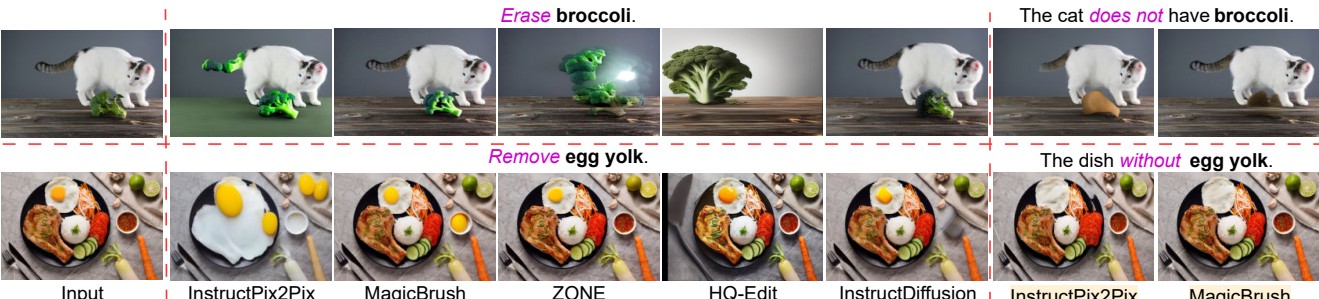

Figure 4. Direct and negative instruction. The fine-tuned InstructPix2Pix and MagicBrush on NeIn's training set are highlighted in the last two columns. Even though our fine-tuned InstructPix2Pix can't perfectly remove the broccoli in the first example, it still recognizes that the broccoli should be removed.

completely alter the content of the query image.

---

**Algorithm 2** Retention Evaluation by OVD

**Input:**
$\mathcal{F}$: samples of NeIn
$\mathcal{T}_o$: original caption from MS-COCO
$\mathcal{T}$: considered model's outputs

**Output:**
$s$: retention score

1: $s := 0$
2: **for** each tuple $(\mathcal{F}^{(i)}, \mathcal{T}_o^{(i)}, \mathcal{T}^{(i)})$ in $(\mathcal{F}, \mathcal{T}_o, \mathcal{T})$ **do**
3:     $\text{list}^1 := [], \text{list}^2 := []$
4:     $\mathcal{O} \leftarrow \text{extractor}(\mathcal{T}_o^{(i)})$     ▷ Original objects in $\mathcal{I}$
5:     $p^1 \leftarrow \text{OVD}(\mathcal{F}^{(i)}, \mathcal{O})$     ▷ Objects are still in $\mathcal{F}^{(i)}$
6:     **for** each $object$ in $p^1$ **do**
        # each object in $p^1$ may overlap with multiple confidence scores; store each object only once
7:        append unique $object$ to $\text{list}^1$
8:     **end for**
9:     $p^2 \leftarrow \text{OVD}(\mathcal{T}^{(i)}, \text{list}^1)$    ▷ Objects in $\mathcal{F}^{(i)}$ & $\mathcal{T}^{(i)}$
10:    **for** each $object$ in $p^2$ **do**
11:      append unique $object$ to $\text{list}^2$
12:    **end for**
13:    $score \leftarrow$ length of $\text{list}^2$ / length of $\text{list}^1$
14:    $s \leftarrow s + score$
15: **end for**
16: $s \leftarrow s / |\mathcal{T}|$
17: **return** $s$

---

InstructDiffusion occasionally handles negative queries effectively, as demonstrated by the third sample and its relatively high scores in removal and retention metrics. We hypothesize that the combination of datasets from various computer vision tasks enhances the model's generalization capabilities, therefore, improving its understanding of negation. HQ-Edit is capable of removing objects specified in the negative prompts from the query image. However, it often significantly alters the overall content of the query image. This suggests that HQ-Edit may not fully understand

the concept of negation; instead, it likely modifies the image content based on the characteristics it learned from its training dataset.

After being fine-tuned on our training set, Instruct-Pix2Pix and MagicBrush are capable of handling negative queries, even with verbs not included in the pre-defined prompts (e.g., "include" in the last query) and nouns absent from the MS-COCO dataset (e.g., "candle" in the third query). We encourage researchers to continue leveraging NeIn for other tasks within the vision-language domain.

# G. Direct Instruction and Negation

Although the "remove" and "erase" instructions appear in text-guided image editing datasets (e.g., MagicBrush contains 7% remove prompts), original image editing models still struggle to remove objects from images. Our fine-tuned models, which leverage negation, provide an alternative yet still natural approach for object removal. Also, in practice, negation is used frequently by human, and we cannot expect users to avoid using negative words with image editing models. We hope that combining NeIn with existing image editing datasets will help models meet user expectations.

# H. Future Directions in Other Vision-Language Tasks

While designed for text-guided image editing, NeIn is potentially suitable for evaluating negation understanding in other vision-language tasks. For distinct tasks, we can use NeIn in appropriate ways to assess models' negation understanding. *For composed image retrieval*, NeIn can be formatted so that its sample, negation prompt, and source image from COCO serve as the query image, query text, and retrieved image, respectively. *For image-text matching*, VLMs can be trained to increase the cosine similarity between the COCO image and the negative instruction, while minimizing the similarity between the NeIn sample and the same caption, thus improving their ability to align images with textual negation.

## I. Photo Credits

- Goldendoodle Dog: instagram/aurixdoodle
- Notre-Dame Cathedral Basilica of Saigon: Indocinatours
- Pizza: Artan's Pizza
- Person and Dog: The Manual
- Mocha Coffee and Book: Mt Zion District Library
- Beach Umbrella: Wirecutter
- Candle: McGee & Co
- Vegetable: The Mediterranean Dish
- Cat and Broccoli: Cats.com
- Vietnamese Broken Rice Dish: Sunhouse