# OpenReview forum: "NeIn: Telling What You Don’t Want"
_thecvf.com/CVPR/2025/Workshop/SyntaGen — SyntaGen 2025 Poster_

### Official Review · Reviewer_ECwe · 2025-03-27
**Review for paper: NeIn: Telling What You Don’t Want**

**Rating:** 7
**Confidence:** 3

**Review:**

Summary:

 Conventional image editing methods are not good at comprehend prompts that contain negative instructions. To address the problem, the paper construct a large dataset via a generation process and a filtering process. The constructed dataset can be used to fine-tune existing VLMs to enhance their ability of understanding negative prompts.

Strengths:

1. The motivation of the paper is clear. Existing training datasets contain few negative prompts can not enable the image editing model of understanding negative instructions.

2. The paper is well-written, Figure. 2 clearly demonstrates the dataset generation process.

3. Experiment results in Table. 3 and Figure. 4 show that the proposed dataset can significantly enhance existing methods.

---

### Official Review · Reviewer_Ggdi · 2025-03-28
**Well-motivated, Large-scale dataset, Good results**

**Rating:** 7
**Confidence:** 3

**Review:**

**Summary**

This paper introduces NeIn, a large-scale dataset aimed at improving and evaluating the ability of vision-language models to understand negative instructions in the setting of image editing. Observing that current models struggle with negation (e.g., "without S"), the authors propose an automated pipeline to generate data pairs: a source image `I` (without object `S`), a negative instruction `Tn` ("no S"), and a target image `F` (synthetically generated by adding `S` to `I`). The evaluation involves tasking models with regenerating `I` given `F` and `Tn`. This paper propose a generation pipeline (using BLIP and InstructPix2Pix/MagicBrush), a filtering pipeline (using BLIP and LLaVA-NeXT), and an evaluation suite including image quality and negation metrics (Removal score, Retention score). Experiments show that current models perform poorly, while models finetuned on NeIn dataset show improvement on the benchmarks.

**Strength**
1. The paper targets a well-motivated and significant limitation of current VLMs – the understanding of negation, a key aspect of human language. Targeting on the image editing makes the problem practically relevant.
2. NeIn is presented as the first large-scale dataset designed for evaluating negation in instruction-based image editing. The scale (over 340k training samples) is quite substantial.
3. The paper introduces an evaluation protocol that considers whether the negated object is removed (Removal Score), whether the rest of the image content is preserved (Retention Score), and standard image quality metrics. This provides a comprehensive validation of model performance.
4. The experiments show the poor performance of existing SOTA image editing models on negative instructions, and the improvements observed after finetuning on NeIn support the dataset's utility.

**Question**

1. What are the cases the model still struggle even after training on NeIn? An analysis of the failure cases of the finetuned models would be insightful.

2. Algorithm 1 & 2 (Threshold α): The threshold α=0.4 is used in both generation (selecting absent objects) and filtering (verifying object presence). Is the pipeline sensitive to this threshold?

---

### Official Review · Reviewer_JccH · 2025-03-28
**Good dataset for an interesting task**

**Rating:** 7
**Confidence:** 4

**Review:**

# Summary
NeIn introduces a comprehensive generated dataset for image editing with negative instructions. One data instance includes a source image, a caption, a selected object for removal, a negative sentence, and a target image. The dataset is built upon several VLMs by using prompts to add an object into image A (target image), resulting in Image B (source image). The bad results are then filtered out by more VLMs. The paper also proposes evaluation protocols for image editing with negative instruction.

# Strengths
- The paper is well-motivated. The challenges are clearly highlighted.
- The results shown are good.  I suggest including your own results in Figure 1.
- The method is explained in detail and can be reimplemented.
- The proposed evaluation metrics are well-designed and can reflect the performance.

# Weaknesses
- No discussion on limitations. This may be out-of-scope, but I wonder how well this would work on more complex prompts like ‘no red cars’ or ‘no running children’.

---

### Decision · Program_Chairs · 2025-03-30

**Decision:**

Accept (Oral)

**Comment:**

The paper received all scores as 7. Overall, the reviewers applauded the paper's strong motivation, clear writing, substantial dataset scale, and convincing results. The Program Chairs checked and agreed to accept the paper.